# Adult axolotls can regenerate original neuronal diversity in response to brain injury

Ryoji Amamoto[1], Violeta Gisselle Lopez Huerta[2], Emi Takahashi[3], Guangping Dai[4], Aaron K Grant[5], Zhanyan Fu[2], Paola Arlotta[1,2]*

[1]Department of Stem Cell and Regenerative Biology, Harvard University, Cambridge, United States; [2]Stanley Center for Psychiatric Research, Broad Institute of MIT and Harvard, Cambridge, United States; [3]Division of Newborn Medicine, Department of Medicine, Boston Children's Hospital, Harvard Medical School, Boston, United States; [4]Athinoula A. Martinos Center for Biomedical Imaging, Massachusetts General Hospital, Harvard Medical School, Charlestown, United States; [5]Department of Radiology, Beth Israel Deaconess Medical Center, Harvard Medical School, Boston, United States

**Abstract** The axolotl can regenerate multiple organs, including the brain. It remains, however, unclear whether neuronal diversity, intricate tissue architecture, and axonal connectivity can be regenerated; yet, this is critical for recovery of function and a central aim of cell replacement strategies in the mammalian central nervous system. Here, we demonstrate that, upon mechanical injury to the adult pallium, axolotls can regenerate several of the populations of neurons present before injury. Notably, regenerated neurons acquire functional electrophysiological traits and respond appropriately to afferent inputs. Despite the ability to regenerate specific, molecularly-defined neuronal subtypes, we also uncovered previously unappreciated limitations by showing that newborn neurons organize within altered tissue architecture and fail to re-establish the long-distance axonal tracts and circuit physiology present before injury. The data provide a direct demonstration that diverse, electrophysiologically functional neurons can be regenerated in axolotls, but challenge prior assumptions of functional brain repair in regenerative species.

**\*For correspondence:**
Paola_Arlotta@harvard.edu

**Competing interests:** The authors declare that no competing interests exist.

## Introduction

Under physiological conditions, the neurogenic capacity of the adult mammalian brain is largely restricted to two neurogenic niches, the subventricular zone of the lateral ventricle, which gives rise to interneurons of the olfactory bulb and the subgranular zone of the dentate gyrus, which generates granule cells of the hippocampus (*Ming and Song, 2011*). Neurons in other brain regions are only generated during embryonic development and are not replaced postnatally.

In contrast to mammals, other vertebrates are endowed with superior capacity to regenerate multiple organs, including parts of the central nervous system (CNS). Among these, urodele amphibians like the axolotl (*Ambystoma mexicanum)* are endowed with the capacity to add new neurons to the brain throughout life (*Maden et al., 2013*) and can regenerate the spinal cord and parts of the brain after mechanical injury (*Burr, 1916*; *Kirsche and Kirsche, 1964a*; *Butler and Ward, 1967*; *Piatt, 1955*). Resection of the middle one-third of one hemisphere, but not the whole hemisphere, in the axolotl telencephalon results in reconstruction of the injured hemisphere to a similar length as the contralateral, uninjured side (*Kirsche and Kirsche, 1964a*; *Kirsche and Kirsche, 1964b*; *Winkelmann and Winkelmann, 1970*). Similarly, after mechanical excision of the newt optic tectum,

**eLife digest** Humans and other mammals have a very limited ability to regenerate new neurons in the brain to replace those that have been injured or damaged. In striking contrast, some animals like fish and salamanders are capable of filling in injured brain regions with new neurons. This is a complex task, as the brain is composed of many different types of neurons that are connected to each other in a highly organized manner across both short and long distances.

The extent to which even the most regenerative species can build new brain regions was not known. Understanding any limitations will help to set realistic expectations for the success of potential treatments that aim to replace neurons in mammals.

Amamoto et al. found that the brain of the axolotl, a species of salamander, could selectively regenerate the specific types of neurons that were damaged. This finding suggests that the brain is able to somehow sense which types of neurons are injured. The new neurons were able to mature into functional neurons, but they were limited in their ability to reconnect to their original, distant target neurons.

More research is now needed to investigate how the axolotl brain recognizes which types of neurons have been damaged. It will also be important to understand which cells respond to the injury to give rise to the new neurons that fill the injury site, and to uncover the molecules that are important for governing this regenerative process.

new tissue fills the space produced by injury (*Okamoto et al., 2007*). Interestingly, in the newt, selective chemical ablation of dopaminergic neurons within a largely intact midbrain triggers regeneration of the ablated pool of neurons (*Berg et al., 2010*; *Parish et al., 2007*). In addition to urodeles, teleost fish have also been extensively studied for their capacity to regenerate the CNS and have led to the identification of some of the molecular signals involved in the regenerative process (*Kizil et al., 2012*). These studies highlight the value of regenerative organisms as models to understand the mechanisms that govern brain regeneration for possible application to the mammalian brain.

However, the mammalian CNS is notoriously complex, and its ability to compute high-level functions, like those of the mammalian cerebral cortex, relies on the presence of a great diversity of neuronal subtypes integrated in specific long-distance and local circuits and working within a defined tissue architecture. Disruption of brain structure, connectivity, and neuronal composition is often associated with behavioral deficits, as observed in models of neurodevelopmental, neuropsychiatric, and neurodegenerative disease. It is therefore likely that functional regeneration of higher-order CNS structures will entail the regeneration of a great diversity of neuronal subtypes, the rebuilding of original connectivity, and the synaptic integration of newborn neurons in the pre-existing tissue. It is not known to what extent even regenerative species can accomplish these complex tasks, beyond their broad ability to generate new neurons and to rebuild gross brain morphology. It remains therefore debated whether any vertebrates are capable of true functional brain regeneration.

Using the adult axolotl pallium as the model system, we have investigated whether a diverse array of neuronal subtypes can regenerate and whether their tissue-level organization, connectivity, and functional properties can also regenerate after mechanical injury.

In contrast to the teleostean pallium, the everted nature of which makes linking distinct regions to their mammalian counterparts difficult (*Northcutt, 2008*), the gross neuroanatomy of the axolotl pallium, organized around two ventricles, shows clear similarities to that of the mammalian telencephalon. In addition, while the evolutionary origin of the mammalian cerebral cortex remains controversial (*Molnár, 2011*), it is likely that the axolotl pallium contains a basic representation of several of the neuronal subtypes found in the mammalian cerebral cortex and thus may serve as a good model for investigating regeneration of neuronal heterogeneity and complex circuit function.

Here, we demonstrate that both pre- and post-metamorphosis adult axolotls are able to regenerate a diversity of neurons upon localized injury to the dorsal pallium. This process occurs through specific regenerative steps that we defined in live animals using non-invasive magnetic resonance imaging (MRI). Strikingly, newborn neurons can acquire mature electrophysiological properties and

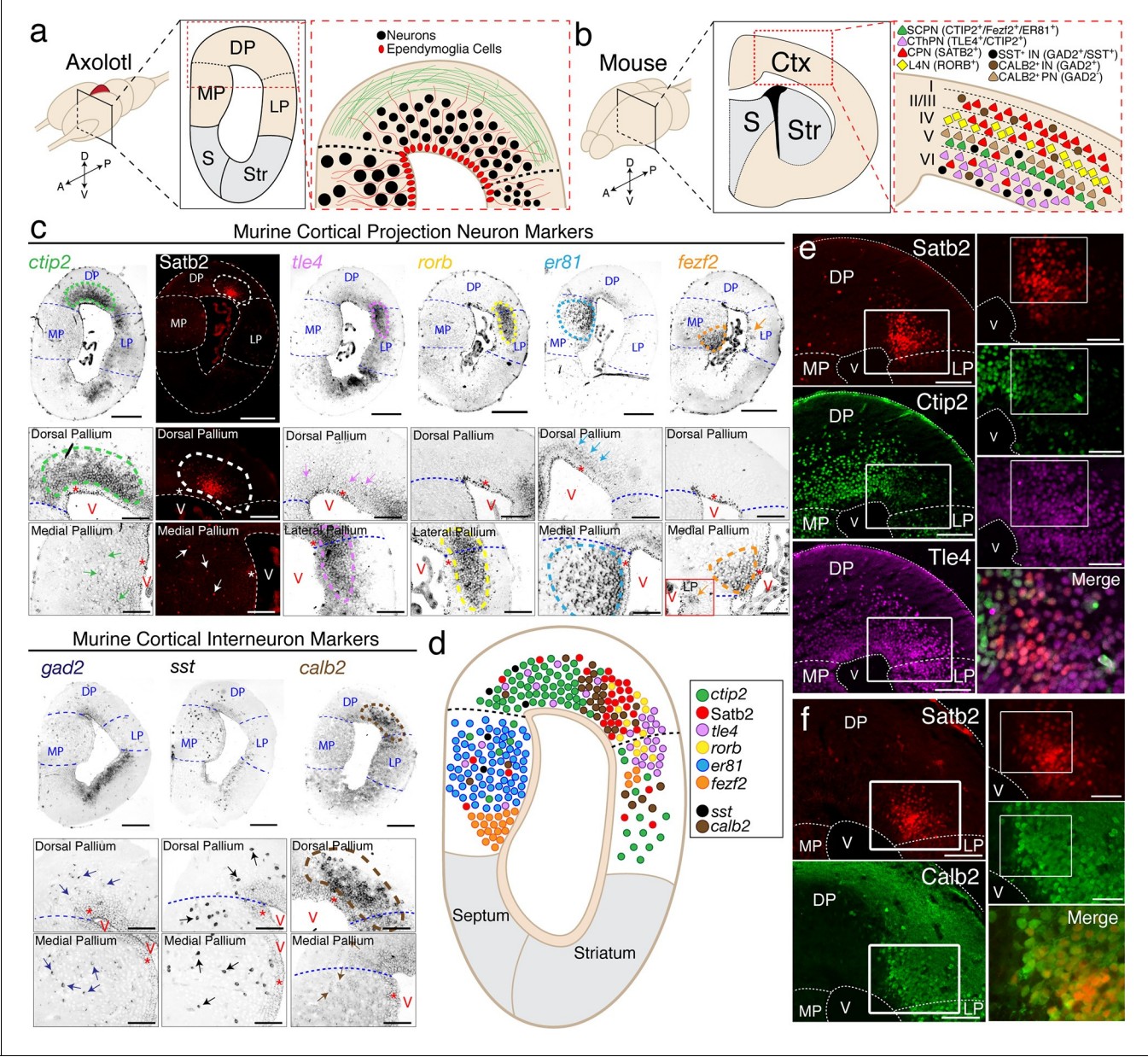

**Figure 1.** The axolotl pallium contains molecularly diverse neuronal subpopulations. (**a**) Schematic representation of the axolotl pallium showing ependymoglia cells (red) lining the ventricle, neurons (black) positioned in the dorsal pallium close to the ventricle, and fiber tracts (green) occupying the region closer to the pia. (**b**) Schematic representation of cortical neuronal subtypes in the mouse neocortex. Major classes of murine cortical projection neuron subtypes as well as selected classes of cortical interneurons are depicted. (**c**) In situ hybridization and immunohistochemistry for selected cortical projection neuron and interneuron markers on coronal sections of the adult axolotl pallium. Insets, high-magnification images of regions of marker expression in the dorsal pallium (top panels) and the medial/lateral pallium (bottom panels). (**d**) Schematic map of the localization of molecularly distinct neuronal subtypes within the adult axolotl pallium. (**e**) Satb2^HI regions in the dorso-lateral pallium are distinctly separate from regions of Ctip2^HI and Tle4^HI expression, showing a molecular boundary. (**f**) Satb2^HI cells largely coexpress Calb2. A, Anterior; P, Posterior; D, Dorsal; V, Ventral; MP, Medial Pallium; DP, Dorsal Pallium; LP, Lateral Pallium; S, Septum; Str, Striatum; Ctx, Cortex; IN, cortical Interneurons; PN, cortical Projection Neurons; SCPN, Subcerebral Projection Neurons; CThPN, CorticoThalamic Projection Neurons; CPN, Callosal Projection Neurons; L4N, Layer 4 Neurons; SST, Somatostatin; V, Ventricle. Scale bars; 500 µm (c, top panels), 200 µm (c, insets; e and f, left panels), 100 µm (e and f, right panels).

The following figure supplement is available for figure 1:

**Figure supplement 1.** Anatomical map of the axolotl pallium.

respond to local afferent inputs. However, they unexpectedly fail to rebuild long-distance circuit and the original tissue architecture.

The data provide the first proof for the precision with which axolotls regenerate a diverse set of neurons, which in turn become electrophysiologically active and receive local afferent inputs. Notably, however, our results also challenge prior assumptions of functional brain regeneration in salamanders by uncovering unappreciated limitations in the capacity of adult axolotls to fully rebuild original long-distance connectivity and tissue organization, a finding that redefines expectations for brain regeneration in mammals.

## Results

### The axolotl pallium contains molecularly diverse neuronal populations

In order to investigate whether axolotls can reconstruct neuronal diversity, we have started by building a molecular map of the neuronal populations present in the axolotl pallium.

The neuronal composition of the axolotl pallium is largely unknown, with the urodele pallium being classically divided into medial pallium (MP), dorsal pallium (DP), and lateral pallium (LP) based on the size and location of cell nuclei, as well as rudimentary connectivity (*Herrick, 1948*; *Kokoros and Northcutt, 1977*; *Westhoff and Roth, 2002*) (*Figure 1a*). We first used Nissl staining of serial sections to build an anatomical atlas of the pallium, which we have subsequently used to precisely match the rostro-caudal location of sections among all animals compared in this study (*Figure 1—figure supplement 1*).

Next, we built a molecular map of neurons present in the pallium. We selected 19 genes that, in the mouse, are known to mark specific subtypes of either excitatory pyramidal neurons or inhibitory interneurons of the cerebral cortex (*Figure 1b*) and performed in situ hybridization on coronal sections of the adult axolotl brain, using riboprobes designed on axolotl cDNA sequences. Among all markers tested, expression of nine genes could be detected reliably in the axolotl pallium: *ctip2, satb2, tle4, rorb, er81, fezf2, gad2, sst,* and *calb2*. In the mouse neocortex, *Fezf2* and *Ctip2* are specifically expressed by subcerebral projection neurons (ScPNs) in layer Vb at high levels and by corticothalamic projection neurons (CThPNs) in layer VI at low levels (*Arlotta et al., 2005*; *Chen et al., 2005a*; *Chen et al., 2005b*; *Molyneaux et al., 2005*). Within these two neuronal populations, *Er81* labels ScPNs (*Yoneshima et al., 2006*), but not CThPNs, which selectively express *Tle4* (*Molyneaux et al., 2007*). Callosal projection neurons (CPNs) across all layers distinctly express *Satb2* and are negative for *Ctip2, Fezf2,* and *Tle4* (*Alcamo et al., 2008*; *Britanova et al., 2008*). Finally, *Rorb* labels locally connected excitatory interneurons of layer IV (*Jabaudon et al., 2012*). Among *Gad2*[+] cortical GABAergic interneurons, *Sst* and *Calb2* mark specific subtypes (*Bartolini et al., 2013*), although *Calb2* has been shown to also be transiently expressed in pyramidal neurons residing in layer Va (*Liu et al., 2014*).

We found that, similarly to the mouse brain, these markers labeled specific regions of the axolotl pallium. *Ctip2* was highly expressed in DP with lower expression in MP and LP; Satb2, detected by immunohistochemistry, was expressed in a restricted region of DP and in scattered cells within MP; *tle4* marked a domain at the border of the dorso-lateral pallium and was expressed at lower levels across DP and MP; *rorb* was expressed in a distinct domain in the dorso-lateral pallium; *er81* was expressed in MP at high levels and in DP at lower levels; *fezf2* was expressed mostly in the ventral MP and in scattered cells within LP (*Figure 1c*). Among the murine cortical interneuron markers, *gad2* was expressed in scattered cells within the dorso-medial pallium and MP, *sst* was expressed in scattered cells across DP and MP, and *calb2* defined a circumscribed domain in DP, as well as sparse cells in MP (*Figure 1c*).

While all genes tested showed unique distributions, the data indicated that markers of subpopulations of mouse pyramidal neurons, which are normally grouped into defined neocortical layers, cluster within distinct domains in the axolotl pallium (*Figure 1d*).

In order to have reliable molecular reference points after regeneration, we selected, for further analysis, four of the nine genes tested - *ctip2, satb2, tle4,* and *calb2* - because of their highly circumscribed and robust profiles of expression within defined regions of the dorsal and dorso-lateral pallium. We used immunohistochemistry to define, at the single cell level, the colocalization of these markers. We found that the dorso-lateral pallium contains Satb2[HI], Ctip2[LO], Tle4[HI] cells and Satb2[LO],

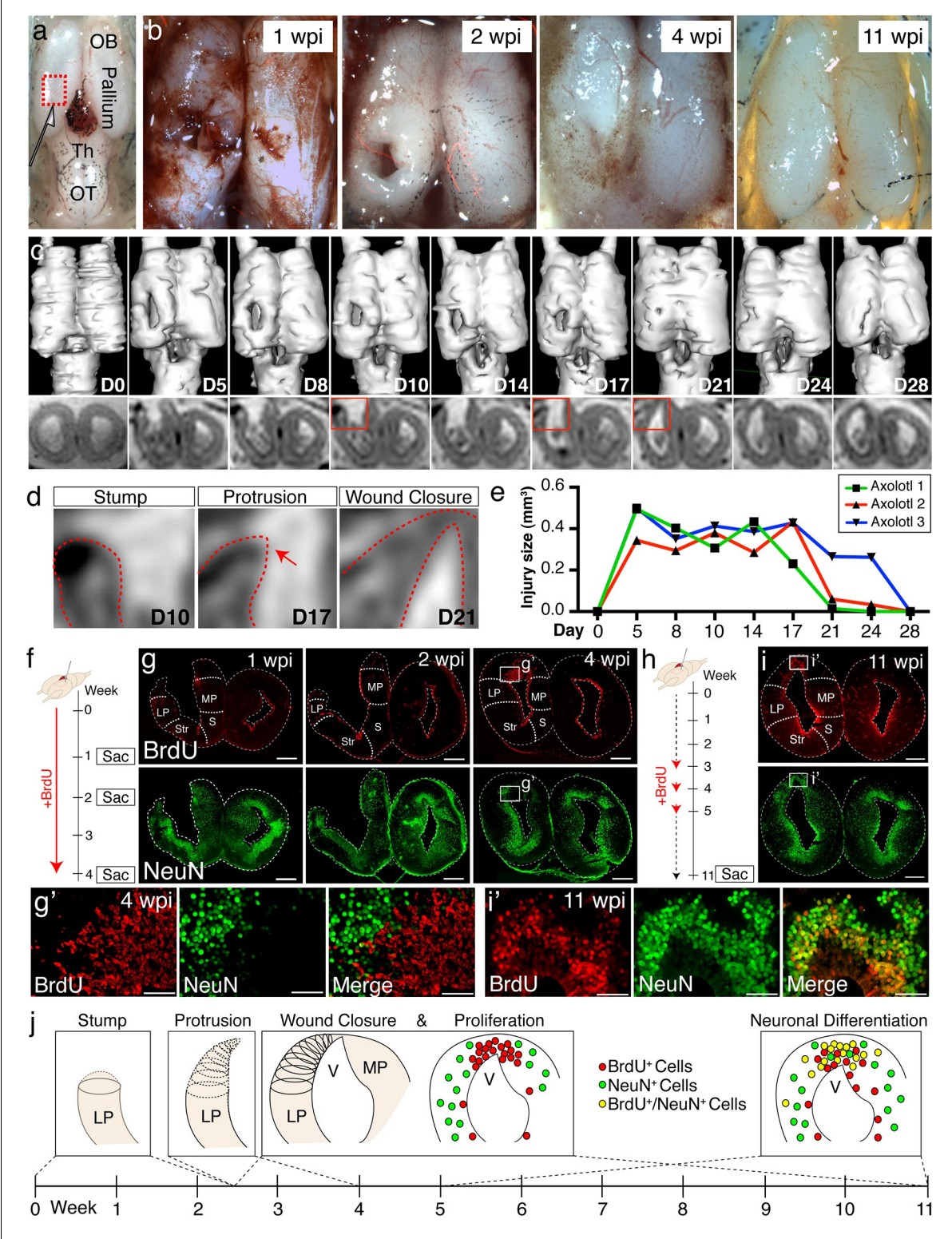

**Figure 2.** Temporal dynamics of successful pallial regeneration after acute mechanical injury. (a) Location of the injury site in the dorsal pallium, left hemisphere. (b) Representative stereoscope photographs of injured brains at 1, 2, 4, and 11wpi. (c) 3D renderings of in vivo MRI images (top panels) and MRI coronal cross-sections within the injury site (bottom panels) of a representative axolotl brain during a 28-day time course post-injury. (d) Enlarged insets (from the red boxes in c) show three distinct stages of wound closure: stump formation, protrusion (red arrow), and closure. Red dotted lines outline the tissue. (e) The sizes of the injury site (in mm³, see Materials and methods) remain unchanged for the first 14 days, but decrease rapidly

*Figure 2 continued on next page*

*Figure 2 continued*

within a subsequent four-day period (n=3). (**f**) Schematic of in vivo BrdU labeling in the early post-injury phase. (**g**) Time course of BrdU and NeuN immunohistochemistry on coronal sections at 1, 2, and 4wpi shows limited cell proliferation within the first 2 weeks and increased proliferation by 4 wpi. Insets, magnified images at 4wpi (panel g'). (**h**) Schematic of in vivo BrdU labeling at mid post-injury phases. (**i**) Immunohistochemistry for BrdU and NeuN on coronal sections at 11wpi shows that a large subset of BrdU$^+$ cells is composed of NeuN$^+$ postmitotic neurons. Insets, magnified images at 11wpi (panel i'). (**j**) Model of the regenerative process during the first 11 weeks after acute mechanical injury. Wound closure proceeds in 3 distinct stages during the first 4 wpi: stump generation, stump protrusion, and wound closure. Only subsequently, newly proliferated BrdU$^+$ cells populate, in large number, the injured region. By 11wpi, newborn neurons are generated. OB, Olfactory Bulb; Th, Thalamus; OT, Optic Tectum; wpi, weeks post injury; D, Day; LP, Lateral Pallium; MP, Medial Pallium; Str, Striatum; S, Septum. Scale bars; 500 μm (g and i), 100 μm (g' and i').

The following figure supplements are available for figure 2:

**Figure supplement 1.** 4wpi proliferative cells express markers of ependymoglia cells.

**Figure supplement 2.** Newborn neurons populate distant, uninjured regions of the brain.

Ctip2$^{HI}$, Tle4$^{HI}$ cells (***Figure 1e***). Additionally, Satb2$^{HI}$ cells in the dorso-lateral pallium also express Calb2 (***Figure 1f***). Cells labeled with these markers also colocalized with NeuN or Hu-C/D, indicating that they are neurons (***Figure 1—figure supplement 1***).

Together, the data provide the first map of molecularly distinct populations of neurons within the adult axolotl pallium to enable injury of defined neuronal groups with fidelity.

## In vivo MRI reveals tissue-level dynamics of regeneration in the axolotl pallium

We established a stereotactic injury model to remove a reproducible portion of the dorsal pallium such that the *ctip2*, *satb2*, *tle4*, and *calb2* domains were ablated (***Figure 2a***). To define the time course of regeneration, we first investigated the overall pallial morphology of fixed whole brain preparations from adult axolotls sacrificed at 1, 2, 4, and 11 weeks post injury (wpi). We found that the wound closes by 4wpi, and is largely undetectable by 11wpi (***Figure 2b***).

The tissue dynamics that accompany brain regeneration in vivo have never been investigated. To gain information on the tissue-level morphological changes that take place over the course of the regenerative process in individual animals and to account for animal-to-animal variability, we performed in vivo MRI of live axolotls undergoing pallial regeneration. This method demonstrated that the wound closure process occurs rapidly, between 2 and 4wpi (***Figure 2c–e*** and ***Video 1***). Notably, the lateral pallium formed a stump, which subsequently thinned out before protruding towards the medial pallium and closing the gap generated by the injury (***Figure 2d***). This indicates that wound healing occurs largely by dynamic tissue remodeling that occurs before cellular proliferation starts.

To assess the cellular dynamics of the first eleven weeks post-injury and to map the time course of neurogenesis, animals were pulsed with Bromodeoxyuridine (BrdU; see Materials and methods). At 4wpi, a large number of BrdU$^+$/NeuN$^-$ cells populated the injured hemisphere, indicating that accumulation of proliferative cells precedes neuronal differentiation (***Figure 2g***). Of note, BrdU$^+$ cells expressed Sox2 and Gfap, two ependymoglia cell markers (***Kirkham et al., 2014***) (***Figure 2—figure supplement 1***). By 11wpi, the majority of BrdU$^+$ cells expressed NeuN (69.5%, n=6 animals)

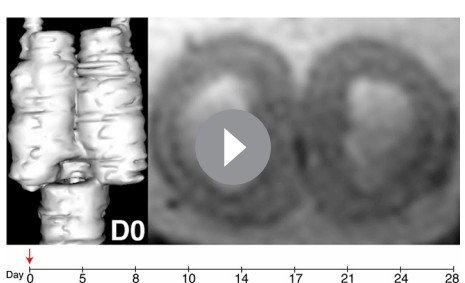

**Video 1.** in vivo MRI reveals temporal dynamics of the first 4 weeks of pallial regeneration. The 3D surface rendering of MRI images (left panel) shows the gradual wound closure process. The raw MRI images (right panel, red dotted lines outline the tissue) show that within the first 2 weeks, the tissue does not change in shape or trajectory. During the next three time points, three stages of wound closure are observed: stump formation, protrusion, and wound closure, which occur rapidly. The time course of regeneration is in days (bottom panel). D, Day; LP, Lateral Pallium; DP, Dorsal Pallium; MP, Medial Pallium.

(*Figure 2i*) indicating that neuronal differentiation occurs between 4 and 11wpi.

Interestingly, we observed an increased number of BrdU$^+$ and PH3$^+$ proliferative cells in regions rostral to the injury site, at the level of the rostral telencephalon, despite the fact that this region was not injured (*Figure 2—figure supplement 2*). To determine whether this distal proliferative response leads to regeneration of new neurons, we quantified the number of BrdU$^+$/NeuN$^+$ cells in the rostral telencephalon at 11wpi in injured and sham controls, in which the skin and skull were opened but the brain was not injured. Quantification revealed that there were significantly more BrdU$^+$/NeuN$^+$ nuclei in the rostral telencephalon of injured animals compared to sham animals, indicating that an injury to the pallium leads to generation of newborn neurons distally (*Figure 2—figure supplement 2*).

Taken together, these data demonstrate that regeneration of the pallium initiates with distinct morphological changes to the injury edge (stump), which thins out and closes the wound. This process precedes rapid proliferation of progenitors and neurogenesis (*Figure 2j*). In addition, the progenitor proliferative response induced by injury is not restricted to the region immediately adjacent to the injury site but induces a more global response in distal uninjured regions.

## Molecular diversity of neuronal subtypes regenerates upon injury to the pallium of both pre- and post-metamorphosis axolotls

To assess the axolotl's ability to regenerate distinct neuronal populations in distinct regions of the pallium, we quantified the number of NeuN$^+$, Ctip2$^+$, Satb2$^+$, Tle4$^+$, and Calb2$^+$ neurons at 11wpi in the dorsal pallium of contralateral, injured, and sham hemispheres. Cells were counted in four consecutive sections, sampled every 300 µm, over a total rostral-caudal length of 900 µm spanning the injury site. The sum of all cells counted in all sections was taken for each sample (*Figure 3a*). From this analysis, we found no significant differences between the number of cells present in the contralateral, injured (regenerated), and sham hemispheres for any of the markers quantified (*Figure 3b–f*). The data indicate that the regenerated pallium contains each neuronal population in numbers comparable to control (contralateral and sham) pallium.

To determine whether there was an increase in the number of newborn neuronal subtypes in the injured pallium compared to controls, we quantified the number of cells that showed colocalization of each neuronal subtype-specific marker with BrdU at 11wpi. This analysis showed that the injury significantly increases the number of newborn neurons within each molecularly defined neuronal subpopulation, compared to controls (n=6 contralateral; n=6 injured; n=4 sham) (*Figure 3g–k* and *Table 1*), indicating that distinct neuronal populations are regenerated upon injury.

These numbers likely underestimate the number of neurons within each population that are newly born due to the difficulty of labeling by BrdU the entirety of the dividing cell pool. It is therefore not possible to conclude whether some of the neurons in the regenerate are newborn (yet not labeled by BrdU) or are endogenous neurons that have migrated from neighboring territories. If migration of neurons into the injured region were a major contributor to the regenerative process, this could cause a decrease in the number of certain neuronal subtypes in regions adjacent to the injury site. We thus quantified the number of neuronal subtypes in MP and LP. We found no significant differences between contralateral, injured, and sham hemispheres for any of the neuronal subtype markers tested (*Figure 3—figure supplement 1*), suggesting, albeit not fully excluding, the possibility that no major waves of migration of endogenous neurons reconstitute neuronal diversity in the regenerated dorsal pallium.

It has been previously reported that the brain regeneration capability of the axolotl decreases upon metamorphosis (*Kirsche et al., 1965*). To address this issue directly, we induced metamorphosis in adult, sexually mature axolotls by administration of L-thyroxine exogenously via immersion (*Page and Voss, 2009*) (*Figure 3—figure supplement 2*). Upon metamorphosis, we injured the dorsal pallium of post-metamorphosis axolotls and found that morphological regeneration takes place within a similar time frame as the pre-metamorphosis axolotls (*Figure 3—figure supplement 2*). At 11wpi, we found that the post-metamorphic injured pallium contains BrdU$^+$/NeuN$^+$, BrdU$^+$/Ctip2$^+$, BrdU$^+$/Satb2$^+$, BrdU$^+$/Tle4$^+$, and BrdU$^+$/Calb2$^+$ nuclei (*Figure 3—figure supplement 2*). Together, these data show that adult pre- and post-metamorphosis axolotls are able to regenerate neuronal subtypes within the dorsal pallium.

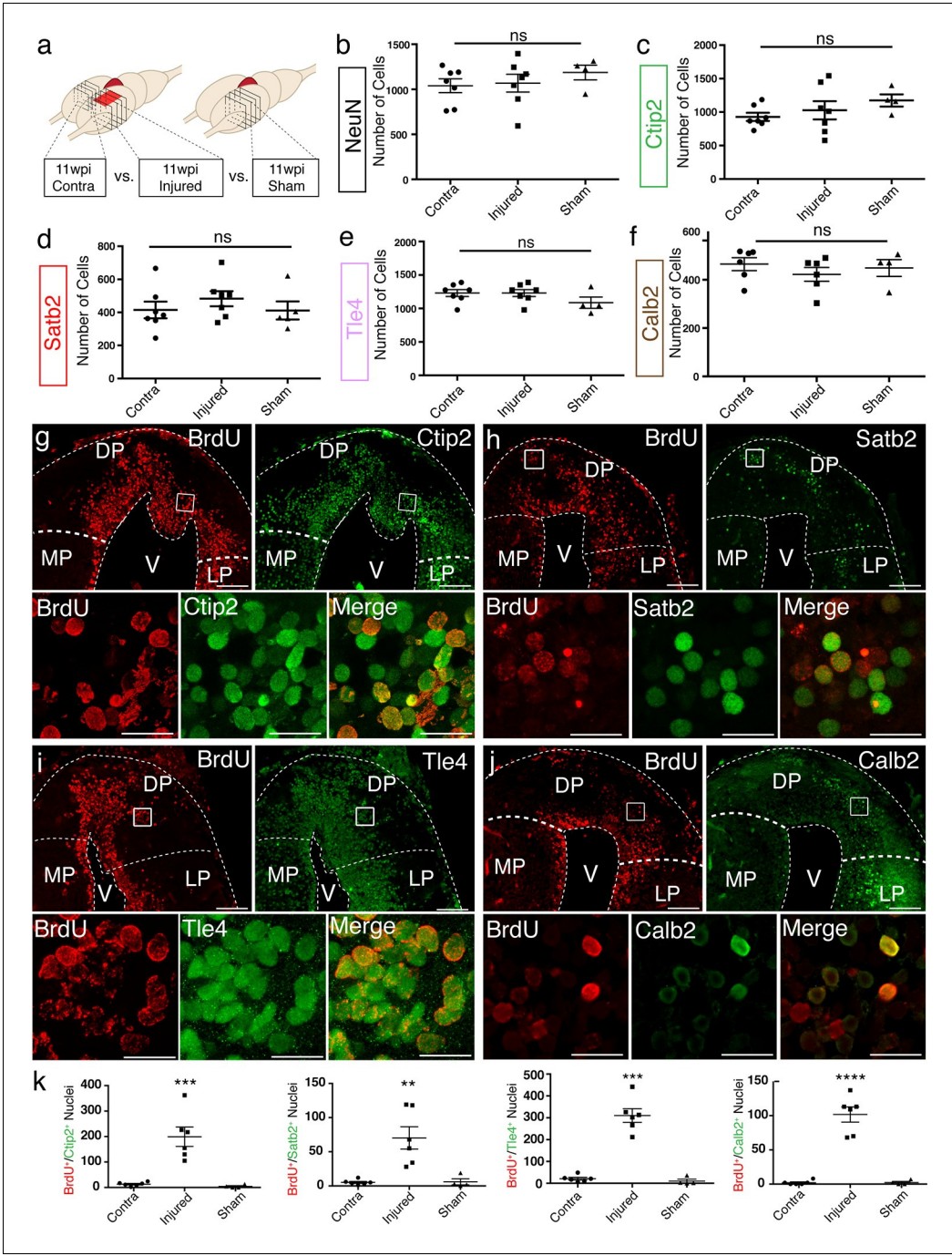

**Figure 3.** Molecularly diverse neuronal subtypes regenerate in the axolotl pallium. (a) Schematic of the sections used for quantification (panels b–f and k). Images from 4 consecutive coronal sections, 300 μm apart, and spanning the injury site from contralateral, injured, and sham hemispheres were used for cell counting. (b–f) Quantification of NeuN[+], Ctip2[+], Satb2[+], Tle4[+], and Calb2[+] nuclei in the region of interest shows no significant differences among contralateral, injured, and sham hemispheres. (g–j) Newborn BrdU[+] cells in injured coronal sections co-express Ctip2, Satb2, Tle4, and Calb2, showing that neuronal subtype diversity is replenished upon regeneration; insets, high-resolution confocal images of co-labeled nuclei. (k) Quantification of BrdU[+]/neuronal subtype-marker[+] nuclei in the dorsal pallium of contralateral, injured, and sham hemispheres. LP, Lateral Pallium; DP, Dorsal Pallium; MP, Medial Pallium; V, Ventricle. Scale Bars; 200 μm (g–j, top panels), 20 μm (g–j, bottom panels). All results are expressed as the mean ± SEM. **p<0.01, ***p<0.001, ****p<0.0001; one-way ANOVA with post-hoc Tukey's multiple comparison test.

*Figure 3 continued on next page*

*Figure 3 continued*

The following figure supplements are available for figure 3:

**Figure supplement 1.** Neuronal subtypes in uninjured regions of the pallium remain unchanged in number at 11wpi.

**Figure supplement 2.** Neuronal diversity in the dorsal pallium is regenerated in post-metamorphic axolotl brain.

## Regenerated neurons in the injured pallium are electrophysiologically active and receive afferent local input

To determine whether the regenerated neurons in the injured pallium mature into functional neurons, we sought to characterize their electrophysiological properties by using whole-cell patch clamp recordings (*Figure 4a*). We compared neurons identified as EdU$^+$ by *post hoc* staining in the dorsal pallium of injured animals (11-15wpi; n=14 neurons) and neurons at matched locations in uninjured control animals (n=9 neurons) (*Figure 4b*). Injected, depolarizing current steps evoked action potentials in all cells recorded in the regenerated pallium (*Figure 4c–d*). We found that regenerated neurons exhibited passive and active membrane properties that were comparable to those of control neurons (*Figure 4e* and *Table 2*). However, in some cases (2/14 neurons), when compared with uninjured neurons, newly regenerated neurons displayed passive and active membrane properties indicative of a slightly immature state, such as a larger $R_m$, a smaller $C_m$, more depolarized resting membrane potential, and broader and shorter action potentials (*Figure 4—figure supplement 1* and data not shown). These results indicate that the regenerated pallium contained neurons of varying maturity, but a large number of regenerated neurons were able to acquire mature electrophysiological traits comparable to endogenous control neurons.

To determine whether the regenerated neurons received synaptic input, we recorded spontaneous postsynaptic currents (sPCSs). We found that all EdU$^+$, regenerated neurons in the injured pallium received postsynaptic currents (*Figure 4f*) and displayed no differences in postsynaptic response compared to control neurons in the uninjured pallium (*Figure 4g*). Most of the spontaneous currents were blocked by the AMPA receptor antagonist, NBQX, suggesting that AMPA receptors were involved in the synaptic transmission (*Figure 4—figure supplement 2*).

To understand the kinetics of large-scale neuronal activity, we performed ex vivo calcium imaging with single-cell resolution (*Figure 4—figure supplement 3*). Measuring the key characteristics of spontaneously generated calcium transients, including the peak amplitude, rise time, and fall time, we detected no significant differences between uninjured (n=5 animals) and injured (11-15wpi; n=4 animals) dorsal pallium neurons (*Figure 4—figure supplement 3* and *Video 2–3*).

Together, these results indicate that the regenerated neurons are able to mature into electrophysiologically functional neurons and to receive and respond locally to afferent inputs from other neurons.

## Original tissue architecture cannot be regenerated in the axolotl pallium

It is unclear whether, beyond generating the diversity of neuronal constituents, the original topography of neurons and fiber tracts in the pallium are also rebuilt upon injury, yet this may be required for proper pallial function.

To quantify the radial organization of neurons in regenerated pallium, we radially subdivided the dorsal pallium into three equally sized bins and quantified the number of NeuN$^+$ neurons in bin 3 (closest to pia) normalized to the area of each individual bin (disorganization index; DI). The sum of DI from four consecutive sections spanning the injury site was taken for each group. In uninjured animals, bin 3 is mostly populated by neuronal processes and does not contain cell bodies. In agreement, we found that virtually no NeuN$^+$ neurons are present within this bin in the contralateral and sham hemispheres. However, in the regenerated pallium at 11wpi, we consistently found neurons in bin 3 (Contra n=7 DI 19.46 ± 3.272; Injured n=7 DI 212.4 ± 65.85; Sham n=5 DI 22.95 ± 8.508), indicating that the radial organization of neurons in the dorsal pallium at 11wpi is disrupted compared to controls (*Figure 5a–b*). To confirm that the observed radial disorganization is not temporary, we

**Table 1.** Quantification of BrdU$^+$/Neuronal Subtype Marker$^+$ nuclei in contralateral, injured (11wpi), and sham hemispheres. The numbers represent the sum of BrdU$^+$/Neuronal Subtype Marker$^+$ nuclei from four consecutive sections spanning the injury site. All results are expressed as the mean ± SEM.

| | Ctip2$^+$/BrdU$^+$ | Satb2$^+$/BrdU$^+$ | Tle4$^+$/BrdU$^+$ | Calb2$^+$/BrdU$^+$ |
|---|---|---|---|---|
| Contralateral (n=6) | 12.83 ± 3.114 | 5.167 ± 1.537 | 21.00 ± 15.35 | 2.000 ± 1.265 |
| 11wpi Injured (n=6) | 199.2 ± 37.93 | 70.33 ± 16.32 | 310.2 ± 76.54 | 101.7 ± 11.10 |
| Sham (n=4) | 4.250 ± 3.065 | 6.000 ± 4.378 | 10.25 ± 17.21 | 2.250 ± 1.652 |

performed the same analysis at 20wpi. Similar to the 11wpi time point, neurons were consistently positioned in bin 3 in the injured hemisphere, but not in the control (Contra n=5 DI 40.87 ± 9.746; Injured n=5 DI 548.6 ± 104.1) (*Figure 5a–b*), indicating that neuronal disorganization remains present at 20wpi and is a stable feature of the regenerated pallium.

The data prompted the question of whether the topography of neuronal subtypes is disorganized in the regenerate. We thus performed immunohistochemistry for Ctip2, Satb2, Tle4, and Calb2 at 20wpi. We found that, while the uninjured contralateral hemisphere showed stereotyped organization, in which neuronal subtypes cluster within distinct domains (*Figure 5c*, top panels), the dorsal pallium of the injured hemisphere contained neuronal subtypes localized outside of their stereotyped domains (*Figure 5c*, bottom panels).

In agreement, immunohistochemistry for Tubb3 in the dorsal pallium of 11wpi contralateral, injured, and sham hemispheres showed that the injured pallium contained scattered neuronal processes that lacked the stereotypical, circumscribed organization along the pia seen in the uninjured pallium (*Figure 5d*).

Taken together, these results indicate that the molecular diversity of neuronal subtypes is rebuilt within an altered pallial architecture.

## Long-distance projections from the dorsal pallium to the olfactory bulb are reduced in the regenerated brain

Given the abnormal tissue architecture of the regenerated pallium, we sought to determine whether regenerated neurons are able to remake long-distance axonal projections to original targets. In amphibians, tracing studies have shown that neurons in the dorsal pallium project to the medial pallium (MP), lateral pallium (LP), and olfactory bulb (OB) (*Westhoff and Roth, 2002*; *Northcutt et al., 1980*; *Neary et al., 1990*). We selected OB-projecting neurons for further analysis because their cell bodies (in DP) are distinctly separated from their axon targets (in OB). We injected red fluorescent microspheres (Retrobeads) in the OB of control, uninjured (n=5) and regenerated axolotls at both 11wpi (n=6) and 42wpi (n=3). All animals were sacrificed one week post-injection. In each sample, we quantified the total number of Retrobead$^+$ neurons present in the dorsal pallium, across four consecutive sections spanning the injury site. We found that a reduced number of neurons were retrogradely traced from the OB in the regenerated versus control axolotls (Uninjured n=5 81.60 ± 13.90 neurons; 11wpi n=6 24.83 ± 5.326 neurons; 42wpi n=3 19.33 ± 2.603 neurons) (*Figure 6a–b*).

Notably, regenerated BrdU$^+$/Retrobead$^+$ neurons accounted for only 10.2% and 6.6% of the total number of OB-projecting neurons located in the DP at 11wpi and 42wpi, respectively, suggesting that the majority of the axons extending from the regenerated pallium to the OB originate from pre-existing BrdU$^-$ neurons that were spared by the injury (*Figure 6c*, arrowheads). Given that the number of Retrobead$^+$ neurons remains significantly reduced at 42wpi, it is unlikely that this is a matter of neuronal maturity or low speed of axon growth. Interestingly, we also found a reduction in the number of Retrobead$^+$ neurons in the lateral pallium, despite the fact that this region was not mechanically injured (Uninjured n=5 118.2 ± 38.96 neurons; 11wpi n=6 21.83 ± 4.629 neurons; 42wpi n=3 29.67 ± 10.35 neurons) (*Figure 6a–b*). However, this was not accompanied by disorganization of the lateral pallium at 11wpi, as detected by NeuN, Tle4, and *rorb* labeling (*Figure 6—figure supplement 1*). Potential severance of the tract that connects the lateral pallium to the olfactory bulb during the dorsal pallium injury may account for this reduction. As expected, we found no significant difference between groups in the retrogradely labeled neurons that project to the olfactory bulb from the medial pallium (*Figure 6a* and data not shown).

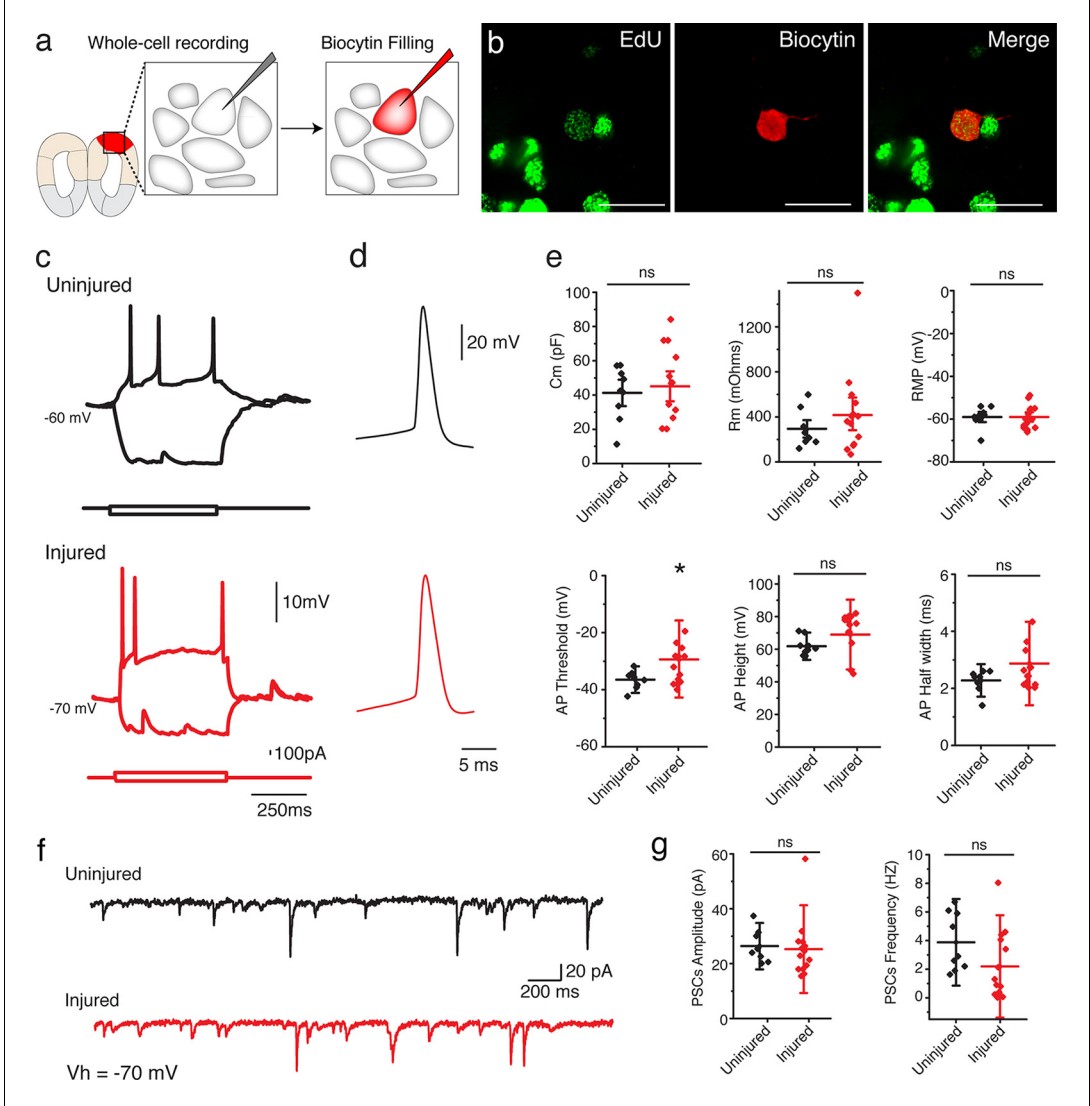

**Figure 4.** Regenerated neurons show electrophysiological features similar to uninjured pallium neurons and receive afferent input. (**a**) Schematic representation of whole-cell patch clamp recording and subsequent biocytin filling. (**b**) Representative images of EdU[+]/Biocytin[+] neurons in the injured pallium. (**c**) Representative traces of the membrane voltage changes in response to depolarizing and hyperpolarizing current steps injections of neurons from uninjured and injured animals. (**d**) Zoom-in of single action potential evoked by a depolarizing current step recorded in neurons of dorsal pallium. (**e**) Summary of passive and active electrophysiological properties: membrane capacitance ($C_m$), membrane resistance ($R_m$), resting membrane potential (RMP), action potential (AP) threshold, AP height and AP half width. (**f**) Sample traces of spontaneous postsynaptic currents (sPSCs) recorded under voltage clamp (Vh = −70 mV). (**g**) Summary of sPSCs features in neurons from uninjured and injured animals. All results are expressed as the mean ± SD. *p<0.05; unpaired, two-tailed Student's t-test.

The following figure supplements are available for figure 4:

**Figure supplement 1.** EdU[+] neurons in the injured pallium show immature electrophysiological properties.

**Figure supplement 2.** Most sPSCs are abolished by administration of AMPA receptor blocker.

**Figure supplement 3.** Spontaneous calcium transients in neurons do not differ between injured and control dorsal pallium.

To further investigate the failure of regenerated neurons to regenerate long-distance projections, we imaged the axon bundles connecting the pallium to the olfactory bulb using diffusion tensor imaging (DTI) in both control and regenerated animals at 11wpi (n=2). In agreement with our

retrograde tracing results, magnetic resonance tractography revealed distinct three-dimensional fiber tracts connecting the pallium and the olfactory bulb in the control, contralateral hemisphere. In striking contrast, we observed significant truncation of the same bundles in the regenerated hemisphere (*Figure 6d*, arrowheads). These results show that regeneration of the pallium does not remake long-distance projections present in the original brain.

To investigate whether regenerated neurons could excite physiological targets that were more closely located, we stimulated the dorsal pallium and performed extracellular field recordings in the lateral and medial pallium. We used ex vivo brain slices of uninjured and injured (11–15wpi) axolotls (*Figure 7a–b*). We found that, in both DP-to-LP and DP-to-MP connections, the amplitude of the pre-synaptic fiber volley (a measure of the number of axons being activated by an electrical stimulus; Negative Peak 1 or NP1) was significantly reduced in the injured pallium compared to controls (DP-to-LP: Uninjured n=5 animals; Injured n=3 animals; DP-to-MP: Uninjured n=4 animals; Injured n=3 animals) (*Figure 7c–d*), indicating that the activity of presynaptic axons is altered upon regeneration due to a change in number or firing properties of axons. In accordance, we also observed that the field population spike (fEPSP) of both LP and MP was significantly decreased in injured axolotls compared to controls upon dorsal pallium stimulation (*Figure 7c–d*), indicating that the MP and LP are not functionally activated to the same extent in the regenerated brain versus controls.

The data indicate that despite the striking ability of axolotl to regenerate a diversity of neuronal subtypes that are electrophysiologically mature and receive local afferent inputs, newborn neurons cannot rebuild long-distance circuit, as was previously assumed. This demonstrates an unknown obstacle to functional brain repair even in a highly regenerative species.

## Discussion

The function of the mammalian brain relies on the processing power of an outstanding diversity of neuronal subtypes, which are integrated into distinct networks necessary for the execution of specific functions. One central goal of regenerative medicine in the CNS is therefore to rebuild not only the original heterogeneity of neurons but also their specific patterns of connectivity within the endogenous tissue.

Prior work suggested that the axolotl might be a good model to understand the mechanisms of complete CNS regeneration because of its capacity to grossly regenerate large portions of the brain when mechanically injured (*Maden et al., 2013*; *Burr, 1916*; *Kirsche and Kirsche, 1964*; *Winkelmann and Winkelmann, 1970*). However, it remains unclear to what extent brain regeneration in the axolotl leads to reformation of functional tissue.

Here, we found that, upon a large mechanical injury to the pallium, the adult axolotl can regenerate the original neuronal diversity. Notably, newborn neurons acquire intrinsic electrophysiological properties and process afferent input in a manner that is indistinguishable from the endogenous neurons in uninjured brains. It is also interesting that this capacity is not lost after metamorphosis, challenging the theory that, in this species, regeneration is partly linked to the maintenance of a paedomorphic state in adulthood (*Kirsche et al., 1965*). These results indicate that, beyond instructing the birth of new neurons, the adult axolotl brain is capable of regenerating a diversity of neurons that in turn are electrophysiologically functional, even when a large region of the brain is removed.

An outstanding question in the field of brain regeneration remains whether, beyond rebuilding cellular complexity and local connectivity, new connections to distant targets can be restored upon brain regeneration. It is similarly unknown whether tissue architecture and neuronal topography can be regenerated. We found that axolotls possess only limited capacity to rebuild original tissue architecture after large deletions of the pallium. In addition, we uncovered an unexpected limitation to

**Table 2.** Parameters of electrophysiological properties of neurons in uninjured and injured brains. All results are expressed as the mean ± SD.

| | Cm (pF) | Rm (mOhms) | RMP (mV) | Threshold (mV) | AP Height (mV) | Half Width (ms) |
|---|---|---|---|---|---|---|
| Uninjured (n=9) | 49.3 ± 16.4 | 294.1 ± 155.8 | -59 ± 4 | -36.5 ± 3 | 61.9 ± 5 | 2.3 ± 0.4 |
| Injured (n=14) | 45.1 ± 21.8 | 426.7 ± 361.6 | -59 ± 5 | -30 ± 6 | 68.7 ± 14.3 | 2.7 ± 0.7 |

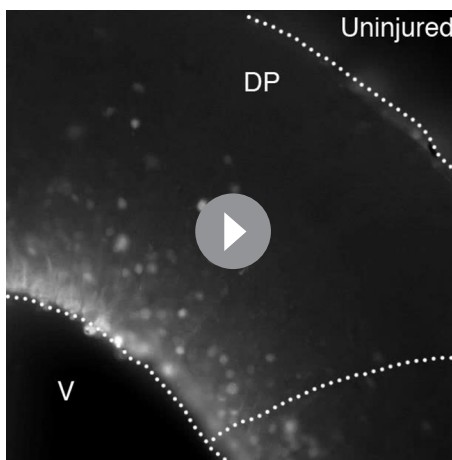

**Video 2.** Calcium imaging in the uninjured dorsal pallium reveals spontaneous calcium transients during homeostasis. The individual cells labeled by Fluo-4 in the control pallium show spontaneous, transient increase in fluorescence. V, ventricle; DP, Dorsal Pallium.

the capacity of newborn neurons to extend axons to original targets in distal brain regions. It is possible that the observed limitations relate to a lack of instructive signals from the environment rather than intrinsic limitations of the regenerated neurons. For example, in order to reach the OB, newborn neurons of the pallium will have to extend axons through uninjured territory that may lack developmental ligands necessary for axon guidance to the OB.

The processes and mechanisms that initially trigger wound closure and subsequently sustain brain repair in regenerative vertebrate species are not known. Using in vivo MRI, we were able to observe, for the first time, the dynamic changes in tissue morphology that occur in live animals over the course of brain regeneration. We found that early steps of wound closure involve the generation of thinner processes from a stump that directionally grow towards each other before fusing. This strategy resembles that observed in the limb, in which closure of the wound by the wound epithelium after amputation is necessary for subsequent regenerative steps and may serve as a source of signals to govern downstream regenerative events (*Mescher, 1976*; *Thornton, 1957*). It is possible that, much like the limb, morphogenetic movements act as a signaling event to trigger cellular proliferation and accumulation of newborn, differentiated cells at the site of injury. In support of this possibility, we did not observe major waves of neuronal migration from regions adjacent to the injury site, and the topography of neurons in these regions remains properly organized upon regeneration. In the future, it will be interesting to determine the cellular source of the newborn neurons. Activated ependymoglia cells are one of the major sources of injury-induced newborn neurons in the newt and zebrafish (*Berg et al., 2010*; *Kroehne et al., 2011*), and these cells may similarly be the main origin of newborn neurons in the axolotl pallium. An alternative, yet not exclusive, possibility is that reprogramming of endogenous differentiated cells may contribute to the generation of new neurons. Such reprogramming events were reported, for example, in the zebrafish retina, where Mueller glia appears capable of generating new neurons upon retinal injury (*Bernardos et al., 2007*; *Fausett and Goldman, 2006*; *Ramachandran et al., 2010*; *Thummel et al., 2008*).

Together, our findings establish the axolotl pallium as a model to further investigate how diverse populations of electrophysiologically functional neurons are mechanistically regenerated. In addition, the results highlight previously unappreciated limits for what vertebrates are able to achieve when faced with the need to regenerate the most complex tissues. This should help define concrete goals and expectations for cellular replacement outcomes in the central nervous system of mammals.

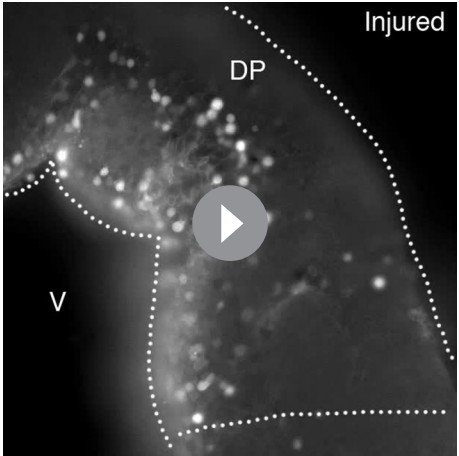

**Video 3.** Calcium imaging in the injured dorsal pallium reveals spontaneous calcium transients. The individual cells labeled by Fluo-4 in the injured pallium show spontaneous, transient increase in fluorescence. V, ventricle; DP, Dorsal Pallium.

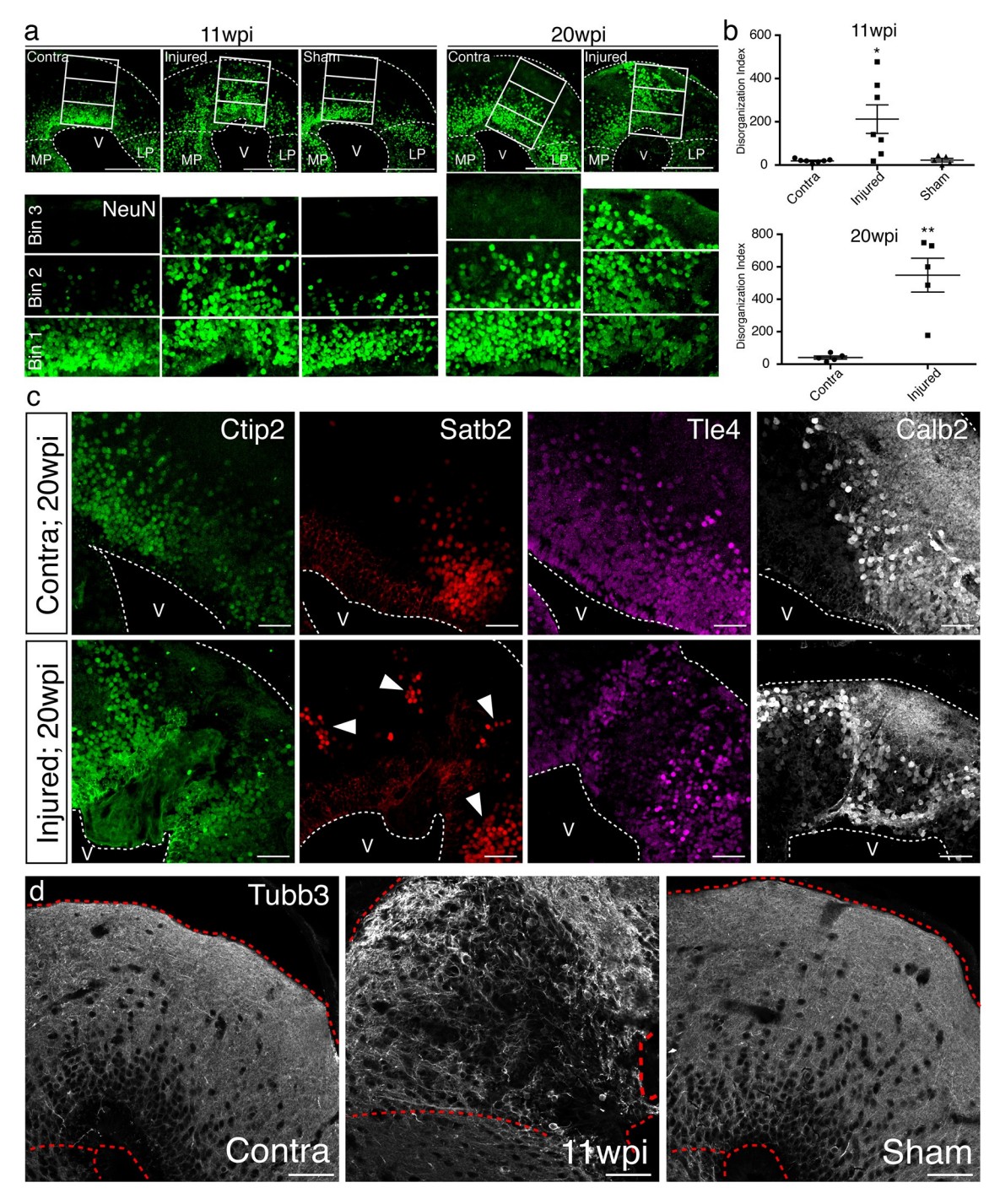

**Figure 5.** Tissue architecture is disrupted in the regenerated pallium. (a) The radial distribution of NeuN[+] cells within the dorsal pallium is altered after regeneration. Representative immunohistochemistry images of coronal sections of contralateral, injured, and sham hemispheres at 11wpi (left) and 20wpi (right). NeuN[+] neurons populate the bin closest to the pia (bin 3) in the injured hemispheres at both time points, in contrast to controls. (b) Quantification of the disorganization index in four consecutive sections of contralateral (11wpi and 20wpi), injured (11wpi and 20wpi), and sham (11wpi) hemispheres. (c) The topography of neuronal subtypes is altered upon regeneration. Representative immunohistochemistry images of coronal sections of contralateral (top panels) and injured (bottom panels) hemispheres at 20wpi. (d) Immunohistochemistry for Tubb3 shows atypical organization of neuronal processes in the 11wpi injured pallium compared to uninjured contralateral and sham controls. LP, Lateral Pallium; MP, Medial Pallium; V, Ventricle. Scale Bars; 400 μm (a), 50 μm (c, d). All results are expressed as the mean ± SEM. *p<0.05, **p<0.01; one-way ANOVA with post-hoc Tukey's multiple comparison test.

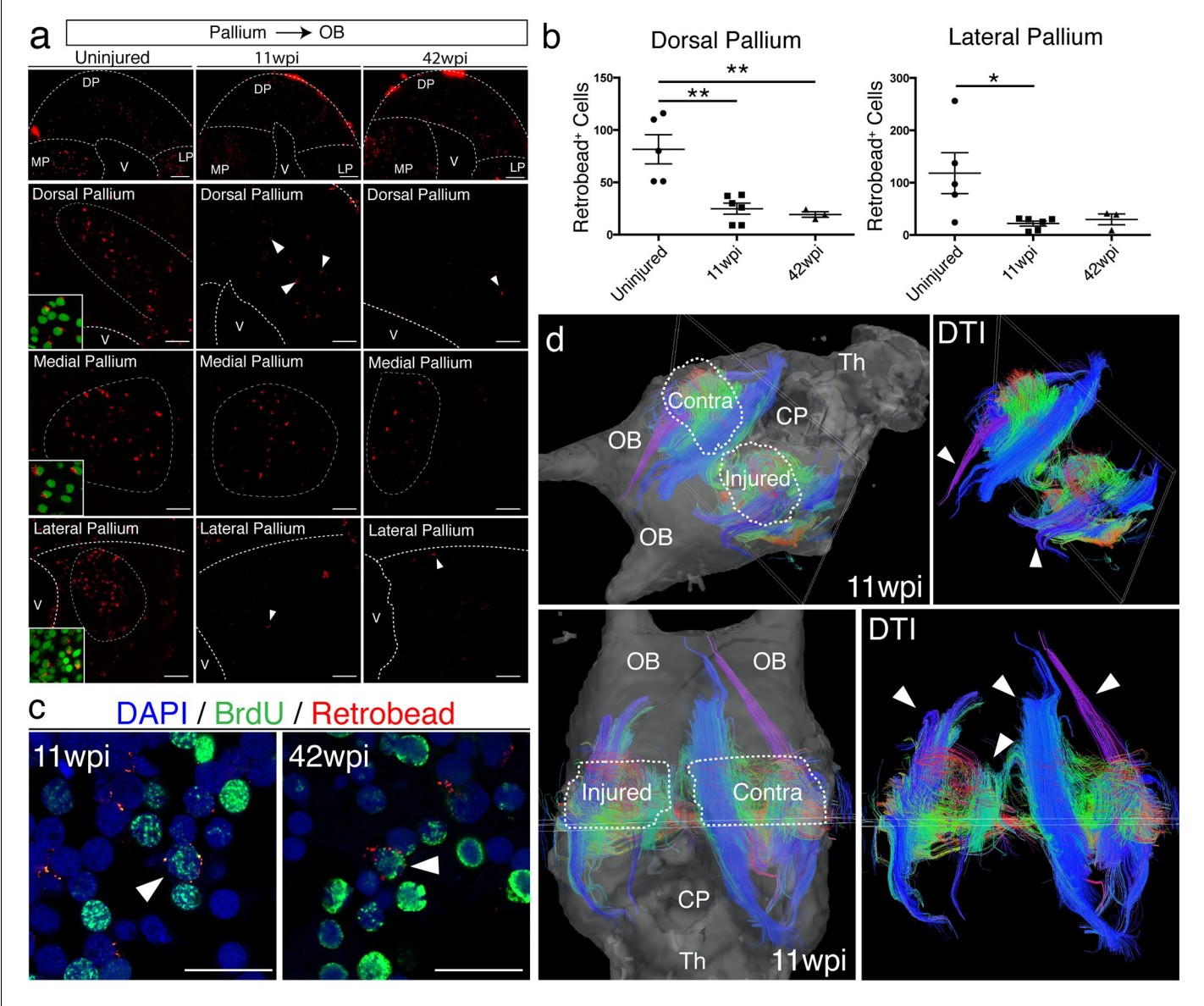

**Figure 6.** Long-distance projections from the regenerated dorsal pallium are significantly reduced. (**a**) Retrograde tracing from the olfactory bulb shows reduced number of labeled cells in the dorsal pallium and lateral pallium at 11wpi (middle panels) and 42wpi (right panels) compared to uninjured control (left panels). Insets, high-magnification images of DAPI (green) and Retrobeads (red). (**b**) Quantification of the number of Retrobead⁺ cells in the dorsal and lateral pallium at different time points after injury. (**c**) Representative immunohistochemistry images of BrdU⁺/Retrobead⁺ cells in the regenerated dorsal pallium at 11wpi (left panel) and 42wpi (right panel). (**d**) ex vivo DTI of 11wpi brains show disruption of fiber tracts (arrowheads) in the injured hemisphere. White dotted lines (left panels) represent regions of interest in injured and contralateral hemispheres. OB, Olfactory Bulb; CP, Choroid Plexus; Th, Thalamus; V, Ventricle; DP, Dorsal Pallium; MP, Medial Pallium; LP, Lateral Pallium. Scale Bars; 50 µm (**b**), 20 µm (**d**). All results are expressed as the mean ± SEM. *p<0.05, **p<0.01; one-way ANOVA with post-hoc Tukey's multiple comparison test.

The following figure supplement is available for figure 6:

**Figure supplement 1.** The neuronal topography of the lateral pallium is not disorganized.

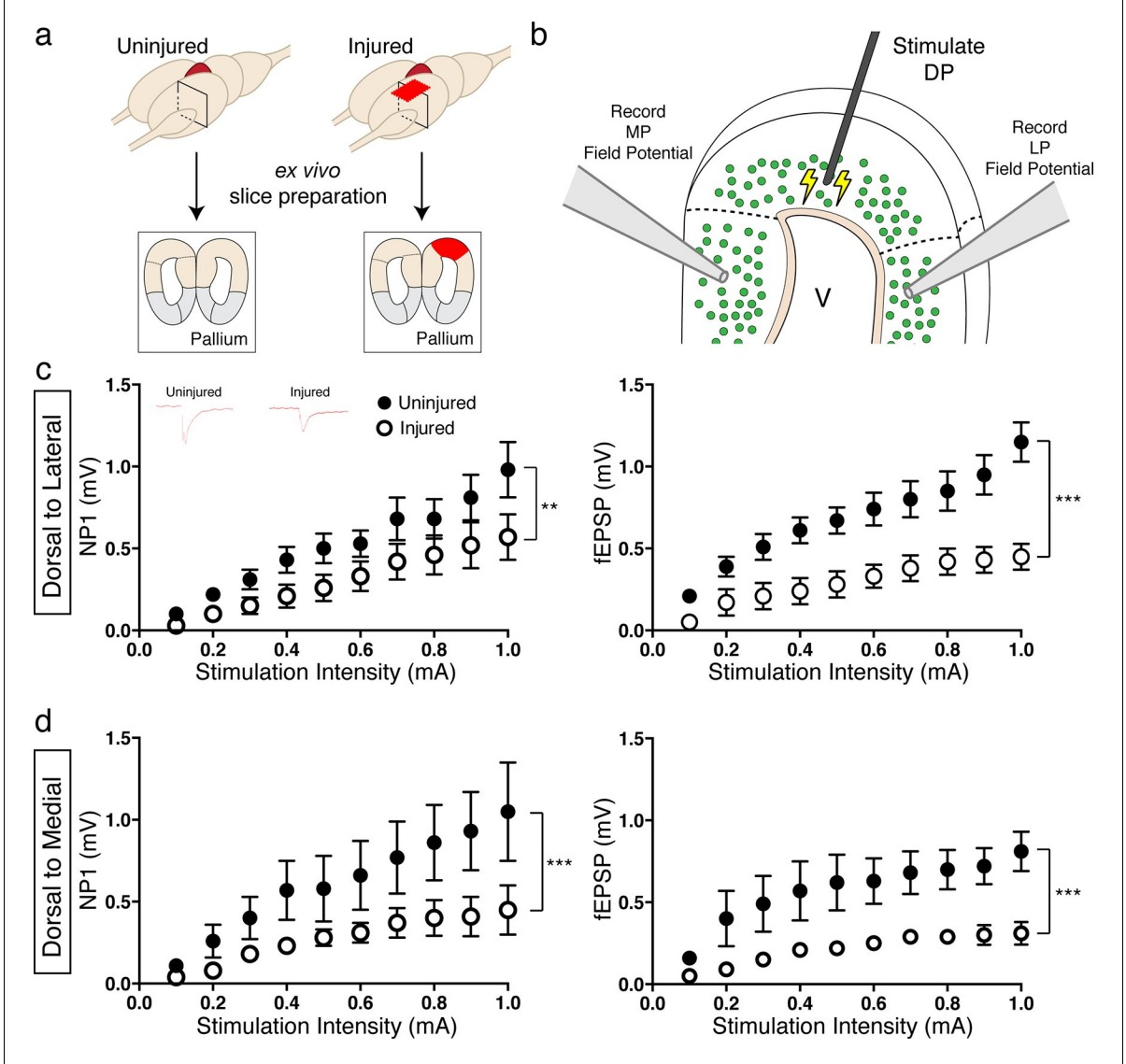

**Figure 7.** Reduced activation of medial and lateral pallium by regenerated neurons in the dorsal pallium. (**a**) Schematic of tissue preparation for slice electrophysiology. (**b**) Schematic of experimental details showing the location of the stimulation (DP) and the field recording (MP and LP). (**c**) Dorsal-lateral NP1 and field pop spike amplitude is decreased in injured pallium (n=15 slices; black circles) compared to control pallium (n=15 slices; white circles) as measured by extracellular field recordings. Inset, representative traces for uninjured and injured pallium recordings. (**d**) Dorsal-medial NP1 and field pop spike amplitude is decreased in injured pallium (n=9 slices; black circles) compared to control pallium (n=9 slices; white circles) as measured by extracellular field recordings. DP, Dorsal Pallium; LP, Lateral Pallium; MP, Medial Pallium; V, Ventricle. All results are expressed as the mean ± SEM. **p<0.005, ***p<0.001; Two-way ANOVA repeated measures with Bonferroni post hoc test.

## Materials and methods

### Animals

Wildtype axolotls (*Ambystoma mexicanum*) were obtained from the Ambystoma Genetic Stock Center (University of Kentucky, Lexington, KY) and were housed in standard conditions at 16°C in Holtfreter's solution. Adult animals, 15–18 cm in length, were used for all experiments. Anesthesia was achieved by immersion in 0.03% ethyl-p-aminobenzoate (benzocaine; Sigma-Aldrich, St. Louis, MO) or 0.1% ethyl 3-aminobenzoate methanesulfonate (tricaine; Sigma-Aldrich). All animal experiments performed were approved by the Harvard University IACUC and were in accordance with institutional and federal guidelines.

## Injury

To access the telencephalon of anesthetized axolotls, two rectangular scalp skin flaps followed by cranial flaps were created dorsally with spring scissors. Incisions outlining a 0.8 mm X 1 mm rectangular injury site were made in the left dorsal pallium with a microknife (Fine Science Tools, Foster City, CA) secured onto a micromanipulator. The caudal-medial corner of the rectangular injury site is defined as 1 mm lateral from and 2 mm rostral from the choroid plexus. The injury site tissue was removed using No.5 forceps. After the injury, the cranial flaps were replaced, and the skins flaps were secured with non-absorbable silk sutures (Myco Medical, Cary, NC). In sham injury, the telencephalon was similarly exposed but the dorsal pallium was left intact.

## BrdU/EdU labeling

250 µl (8 mg/ml in PBS) of BrdU (Sigma-Aldrich) or EdU (Thermo Fisher Scientific, Waltham, MA) was injected intraperitoneally. For animals sacrificed at 4wpi, BrdU/EdU was injected three times a week while for animals sacrificed at 11wpi, BrdU/EdU was injected once at 3, 4, and 5wpi due to the toxic effect of more frequent regimens of BrdU/EdU administration. BrdU incorporation was analyzed by immunohistochemical staining (see below). EdU incorporation was detected with the Click-it EdU Alexa Fluor 488 Imaging Kit (Thermo Fisher Scientific).

## Immunohistochemistry, in situ hybridization, and Nissl staining

Uninjured brains were dissected and fixed overnight in 4% paraformaldehyde (PFA) at 4°C. Injured and sham-injured brains were fixed overnight with the cranium attached, dissected from the cranium, and re-fixed overnight in 4% PFA at 4°C. The extra fixation step is necessary to preserve the morphological integrity of the injured tissue. After fixation, the brains were equilibrated in 30% sucrose solution at 4°C, embedded in OCT (Sakura Finetek, Torrance, CA), and cryosectioned coronally at 30 µm thickness. 10 series (120 sections) were collected from each brain onto Vistavision microscope slides (VWR, Radnor, PA) and stored at -80°C. For BrdU detection, sections were incubated in 2M HCl for 20 min at 37°C, before incubation in blocking buffer (0.3% BSA, 8% serum, 0.3% Triton X-100) for 2 hr at room temperature (RT). Sections were incubated overnight at 4°C in primary antibodies, which were diluted in blocking buffer. Primary antibodies used in this study were as follows: rat anti-CTIP2 antibody 1:100 (Abcam, Cambridge, MA), mouse anti-SATB2 1:50 (Abcam), rabbit anti-TLE4 1:3000 (gift of Stefano Stifani), rabbit anti-CALB2 1:300 (Swant, Marly, Switzerland), rat anti-BrdU 1:300 (Accurate Chemical, Westbury, NY), mouse anti-BrdU 1:300 (BD Biosciences, San Jose, CA), mouse anti-NeuN 1:300 (Millipore, Billerica, MA), rabbit anti-Hu-C/D 1:300 (Genetex, Irvine, CA), rabbit anti-PH3 1:300 (Millipore), rabbit anti-Sox2 1:100 (Abcam), mouse anti-GFAP 1:300 (Sigma-Aldrich), mouse anti-Tubb3 1:300 (Covance, Princeton, NJ). Sections were incubated in appropriate secondary antibodies from the Molecular Probes Alexa Fluor series (Thermo Fisher Scientific) at 1:750 dilutions for 2 hr at RT. After washing, sections were incubated in DAPI 1:50,000 (Thermo Fisher Scientific) for 10 min at RT. Sections were coverslipped with Fluoromount-G (SouthernBiotech, Birmingham, AL) before imaging. Tissue sections were imaged using a Nikon 90i fluorescence microscope equipped with a Retiga Exi camera (Q-Imaging, Surrey, Canada) and analyzed with Volocity image analysis software v4.0.1 (PerkinElmer, Waltham, MA). Confocal images were acquired with Zeiss LSM 700 confocal microscope and analyzed with the ZEN Black software.

In situ hybridization was performed as described previously (*Lodato et al., 2014*). Riboprobes were generated as previously described (*Arlotta et al., 2005*). Riboprobes for in situ hybridization were generated from axolotl cDNA using the following primers:

*fezf2,* F: AAAGCGACAGCAAACTCAGC R: GTTTCCTTTCTGGTGGAAGC;
*ctip2,* F: CCGTTCAGTCTTTTGCGAAT R: TAGCTGCCTTCCATCAATCC;
*tle4,* F: AACAAACAGGCGGAAATTGT R: CTCTAAAGCTTGCCGATGGA;
*rorb,* F: GAAATTTGGCAGGATGTCCA R: ACGTCTCCAGGTGGGATTTA;
*er81,* F: ATGACCAGCAAGTGCCTTTT R: AGGTTTGACTGCTGGCATTG;
*sst,* F: CAGACAAGCAAGCAGCAGAG R: TCTGTGGGAACTGCAGAGTG;
*calb2,* F: GGTCGAGTGCCGAGTTTATG R: TTCACCCTCTCCCCAAATAA;
*gad2,* F: GCGAGCAAAGGGTACAGAAG R: GGCAATACATTTTTCCTTCAAAA

Nissl staining was performed as previously described (*Macklis, 1993*).

All primary data from immunohistochemistry and in situ hybridization experiments were repeated at least three times and analyzed by one investigator, then confirmed by a second, independent investigator who was blinded to experimental conditions.

## Cell quantification and statistical analysis

For quantification of proliferating cells and newborn neurons, anatomically matched sections were processed to detect BrdU/PH3 or BrdU/NeuN, respectively. For quantification of the disorganization index, boxes of 400 µm in width and spanning the thickness of the pallium were superimposed at matched locations on each section and divided into three equally sized bins. For quantification of Retrobead$^+$ cells, anatomically matched sections were processed to detect Retrobead$^+$/DAPI$^+$ cells. All counts were performed by an investigator who was blinded to the condition. Unpaired, two-tailed t-test, one-way ANOVA, or two-way ANOVA was used for statistical analysis.

## Induction of metamorphosis

Adult axolotls of 15–18 cm in length were induced to undergo metamorphosis as described previously (*Page and Voss, 2009*). The animals were treated with L-thyroxine (50 nM, Sigma-Aldrich) for 6 months while kept in 50% water, 50% land habitat. Upon complete gill absorption and exhibition of land walking capabilities, they were transferred to a peat moss habitat and switched to a cricket diet. No surgery was performed before animals were able to feed independently.

## Retrograde tracing

For retrograde tracing, Retrobeads (Lumafluor, Durham, NC) were injected (69 nl/injection, 3 injections per site) into regions of interest in the adult axolotl brain in vivo. One week after injection, animals were sacrificed and the brains were processed as described above.

## Whole-cell patch clamp recordings

The dorsal pallium and individual neurons were visualized and identified with a microscope equipped with Nomarski optics and infrared illumination (BX-51WI, Olympus, Shinjuku, Japan). Whole-cell patch clamp recordings were obtained from axolotl pallium of injured and uninjured animals using recording pipettes (Glass type 8250, King Precision Glass, Claremont, CA) pulled in a horizontal pipette puller (P-87, Sutter Instruments, Novato, CA) to a resistance of 3–4 MΩ, and filled with internal solution containing (in mM): 117.0 K-gluconate, 13.0 KCl, 1.0, MgCl2, 0.07 CaCl2, 0.1 EGTA, 10.0 HEPES, 2.0 Na-ATP, 0.4 Na-GTP, and 0.3% biocytin, pH adjusted to 7.3 with KOH and osmolarity adjusted to 298–300 mOsm with 15 mM K$_2$SO4.

For data acquisition and analysis, a Multiclamp 700B amplifier (Molecular Devices Corporation, Sunnyvale, CA) and digidata 1440A were used to acquire whole cell signals. The signals were acquired at 20 KHz and filter at 2 KHz. Conventional characterization of neurons was made in voltage and current clamp configurations. Access resistances were continuously monitored and experiments with changes over 20% were aborted. Analyses were performed using Origin (version 8.6, Microcal, Malvern, UK) and MiniAnalysis (Synaptosoft, Decatur, GA).

## Extracellular field recording

Ex vivo 300 µm thick slices were prepared by sectioning uninjured and injured (11–15wpi) brains on the vibratome. The extracellular field recording was performed as described (*Peça et al., 2011*). Briefly, a slice was placed into recording chamber (Warner Instruments, Hamden, CT) and constantly perfused with oxygenated aCSF at room temperature at a speed of 2.0 ml/min. A platinum iridium concentric bipolar electrode (FHC) was positioned on the dorsal pallium to stimulate the neurons either in uninjured control or injured slice. A borosilicate glass recording electrodes filled with 2M NaCl was placed onto the lateral or medial pallium approximately 600 µm away from the stimulating electrode. Dorsal-lateral or dorsal-medial field population spikes were elicited by delivery step depolarization (0.15 ms duration with 0.1 mA intensity at a frequency of 0.1 Hz). Stable baseline response of pop spike for at least 5 min from individual slice was ensured before moving to input-output assay. Input-output curves were determined for both the negative peak 1 (NP1; fiber volley) and pop spike amplitude by delivery of three consecutive stimulations from 0 to 1 mA with 0.1 mA

increments. Recordings were performed at room temperature and data were sampled using pCLAMP 10 software (Molecular Devices).

## Calcium imaging

Ex vivo 300 μm thick slices were prepared by sectioning uninjured and injured (11–15wpi) brains. Staining of the slices in Fluo-4 (Thermo Fisher Scientific) was performed in accordance to protocol (*Dawitz et al., 2011*). Slices were imaged on Zeiss Cell Observer for 10 min with a frame rate of 3 Hz. The resulting movie was analyzed using the FluoroSNAPP software (*Patel et al., 2015*).

## In vivo magnetic resonance imaging

Animals for imaging were anesthetized by immersion in 0.1% tricaine (Western Chemical, Ferndale, WA). Once anesthesia was achieved, animals were transferred to an MRI-compatible bed where anesthesia was maintained by partial immersion in tricaine with exposed skin covered by a damp towel.

Animals were imaged using a 9.4T horizontal bore animal imaging system (Biospec 94/20, Bruker Corporation, Billerica, MA). A circular, 20 mm receive-only surface coil was placed over the head. The animals were then situated with their heads near the isocenter of the magnet within an 84 mm quadrature transmit/receive volume coil. $T_2$-weighted images were acquired using a RARE (rapid acquisition with relaxation enhancement) sequence (*Hennig et al., 1986*). All images had RARE factor 8 and echo time TE=33 ms, and a sufficient number of slices were acquired to cover the entire brain. Sagittal images were acquired with repetition time TR=2181 ms, 1 mm slice thickness, 158 μm × 188 μm in-plane resolution, and 1 signal average. Axial images were acquired with TR=2000 ms, 500 μm slice thickness, 150 μm × 150 μm in-plane resolution, and 16 signal averages. Coronal images were acquired with TR=3817 ms, 500 μm slice thickness, 158 μm × 188 μm in-plane resolution, and 4 signal averages.

3D surface rendering of the MRI images were created using OsiriX (*Rosset et al., 2004*). Regions of interest (ROI) were manually drawn on consecutive coronal $T_2$-weighted images to outline the contours of the brain. The pixel values within and outside of the ROIs were changed to 65,000 and -3024, respectively. 3D surface rendering was performed with these settings: highest resolution, 100 iterations, and 1000 pixel values. The sizes of the injury were determined by manually drawing ROIs for regions that are lacking continuous closure of the ventricle. The volumes of the ROIs were calculated using the Compute Volume algorithm.

## Ex vivo diffusion tensor imaging

The brains of anesthetized axolotls were extracted and fixed in 4% PFA containing 1 mM gadolinium (Gd-DTPA) MRI contrast agent to reduce the T1 relaxation time while ensuring that enough $T_2$-weighted signal remained. For MR image acquisition, the brains were placed in formalin. Brains were scanned on a 9.4T Bruker Biospec MR system. The pulse sequence used for image acquisition was a 3D diffusion-weighted spin-echo echo-planar imaging sequence, TR=330 msec, TE=31.23 msec, number of segments 4, with an imaging matrix of 232 × 96 × 96 pixels. Spatial resolution was 50 × 50 × 50 μm. Sixty diffusion-weighted measurements and one non-diffusion weighted (b=0) measurement were acquired, corresponding to a cubic lattice in Q-space at b = 12,000 sec/mm$^2$ with δ = 12.0 msec, Δ = 24.2 msec, with 8 averages. The total acquisition time was 17 hr and 11 min for each imaging session.

Diffusion Toolkit and TrackVis (http://trackvis.org) were used to reconstruct and visualize tractography pathways. Trajectories were propagated by consistently pursuing the orientation vector of least curvature. The color-coding of tractography pathways is based on a standard RGB code, applied to the vector between the end-points of each fiber (red: left-right, green: dorsal-ventral, and blue: anterior-posterior directions). We terminated tracking when the angle between two consecutive orientation vectors was greater than the given threshold (60°) for each specimen. ROIs were placed at the injury site and anatomically matched region in the contralateral hemisphere. Representative 3D fiber tracts that pass through the ROIs and a single coronal plane are shown.

## Acknowledgements

We would like to thank Simona Lodato and Hsu-Hsin Chen for insightful advice and assistance with the manuscript, members of the Arlotta and Melton laboratories for experimental support and discussions, in particular Dennis Sun and Stephanie Tsai. We are grateful to Jessica Whited (Harvard Medical School) for advice and sharing of reagents, Zachary Trayes-Gibson for outstanding technical support, Stefano Stifani (McGill University) for the generous gift of the TLE4 antibody, and Connie Cepko (Harvard Medical School), John Rinn (Harvard University) and Cliff Tabin (Harvard Medical School) for insightful discussions as members of RA's dissertation advisory committee. We thank the genome resources of the Sal-Site (supported by NIH R24OD010435) for the axolotl cDNA sequences. PA is a New York Stem Cell Foundation Robertson Investigator.

## Additional information

### Funding

| Funder | Grant reference number | Author |
|---|---|---|
| National Institute of Neurological Disorders and Stroke | F31NS089336 | Ryoji Amamoto |
| National Institutes of Health | R01HD078561 | Emi Takahashi |
| National Institutes of Health | R21HD069001 | Emi Takahashi |
| Brain and Behavior Research Foundation | NARSAD Young Investigator Award | Zhanyan Fu |
| Stanley Center for Psychiatric Research at Broad Institute of MIT and Harvard | 6045290 | Zhanyan Fu |
| New York Stem Cell Foundation | | Paola Arlotta |
| Harvard Stem Cell Institute | | Paola Arlotta |
| National Institutes of Health | NS062849 | Paola Arlotta |
| National Institutes of Health | NS078164 | Paola Arlotta |

The funders had no role in study design, data collection and interpretation, or the decision to submit the work for publication.

### Author contributions

RA, Conceived the work, Designed the experiments, Performed the majority of the experiments, Wrote the manuscript; VGLH, ZF, Performed and analyzed the electrophysiological experiments; ET, Performed and analyzed the ex vivo DTI experiments; GD, Performed the ex vivo DTI experiments; AKG, Performed and analyzed the in vivo MRI experiments; PA, Conceived the work, Designed the experiments, Wrote the manuscript, Supervised all aspects of the project

### Author ORCIDs

Paola Arlotta, http://orcid.org/0000-0003-2184-2277

### Ethics

Animal experimentation: All animal experiments performed were approved by the institutional animal care and use committee protocol (#12-14) of Harvard University. All surgeries were performed under benzocaine or tricaine anesthesia to minimize pain and suffering during the procedures.

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
