## [Decision Letter]

Thank you for submitting your article "Adult axolotls can regenerate complex neuronal diversity and local circuit in response to brain injury" for consideration by *eLife*. Your article has been reviewed by two peer reviewers, and the evaluation has been overseen by K VijayRaghavan as the Reviewing and Senior Editor.

The reviewers have discussed the reviews with one another and the Reviewing Editor has drafted this decision to help you prepare a revised submission.

The following individuals involved in review of your submission have agreed to reveal their identity: Heinrich Reichert (peer reviewer).

Summary:

Adult salamanders have the apparent ability to regenerate parts of the nervous system, however the precise nature and extent of this regenerative capability is not well characterized. In this report, the authors investigate the extent to which the brain, specifically the pallium, of the adult axolotl can regenerate its neuronal types, their tissue organization, electrical properties and connectivity in response to mechanical injury. For this, they first characterize the diversity of the neurons in the normal pallium based on the in situ expression of a set of nine molecular markers known to label pyramidal neurons or interneurons in mouse. As in mouse, these markers label neurons in a regionalized manner in the axolotl pallium implying a map-like organization of diverse, molecularly defined neuron types in the pallium.

Next, the authors study the regeneration response to injury of the dorsal pallium at the overall anatomical level, and show that a morphological "wound-closure" -like response takes place in the pallium before proliferation of new cells is initiated. Molecular marker studies using 4 of the previously mapped labels together with cell counts in the regenerated pallium indicate that at least a subset of the molecularly distinct neuronal subtypes in the pallium are regenerated and that their number is comparable to that of controls. Further electrophysiological studies indicate that the regenerated neurons have passive and active membrane properties and also manifest spontaneous postsynaptic currents comparable to controls. Thus in these molecular and physiological respects, the regenerated neurons resemble control neurons.

In contrast, the authors find that in terms of tissue architecture, formation of long distance projections and ability to activate target structures in the pallium, the regenerated neurons are markedly defective. Thus, the radial organization of neurons in the dorsal pallium is perturbed, molecularly defined neuronal types are mislocalized, and neuronal fibers manifest misprojection phenotypes. Moreover, long distance projections from the regenerated dorsal pallium to the olfactory bulb are strongly reduced (10% of control values) and the field population spikes in lateral and medial pallium were significantly decreased implying lack of input to these regions.

Taken together, these findings show that the extent of brain regeneration in axolotl is more limited than previously thought. Diverse neuronal types are generated and have relatively normal electrical properties. However, these newly generated neurons do not form the correct tissue architecture nor do they form the appropriate projections to other parts of the brain implying that the reformation of functional brain circuitry is strongly impaired.

These findings have been thoroughly investigated, well documented and clearly presented. They will be of great value to the field as they describe not only the extent to which regeneration can occur in a higher brain centre of a vertebrate (generation of diverse neuron types with normal electrophysiological properties), but equally importantly, also what its limitations are (inability to form functional connections).

Concerns:

The authors interpret the existence of postsynaptic currents in regenerated neurons to indicate that these neurons are integrated into local circuits. This is a clear overinterpretation. Receiving synaptic input not circuit integration. There is no evidence at all that the regenerated neurons make synaptic connections to any neurons in the brain, local or long distance. Nor is there any evidence that the synaptic input to the regenerated neurons in appropriate for any type of circuit function. The new neurons get synaptic input and that is all that the authors show. This over-interpretation should be addressed in the revised manuscript.

Specific comments:

It is interesting to note that the process of regeneration first involves large-scale tissue remodelling akin to 'wound repair' before cell proliferation and differentiation. The authors also show that there is no change in the numbers of cells in the lateral pallium and no cell division either. Does this process then result in re/disorganisation of the lateral pallium? Tracings from the olfactory bulb suggest that this might be the case. Could the authors please comment on this?

Have the authors at all pursued what the progenitors of the regenerating neurons might be? Are there stem cells that are reactivated upon injury? Or is there a process akin to transdifferentiation that contributes to the regeneration? These aspects could also be discussed in the revised manuscript.

---

## [Author Response]

Concerns:

The authors interpret the existence of postsynaptic currents in regenerated neurons to indicate that these neurons are integrated into local circuits. This is a clear overinterpretation. Receiving synaptic input not circuit integration. There is no evidence at all that the regenerated neurons make synaptic connections to any neurons in the brain, local or long distance. Nor is there any evidence that the synaptic input to the regenerated neurons in appropriate for any type of circuit function. The new neurons get synaptic input and that is all that the authors show. This over-interpretation should be addressed in the revised manuscript.

We thank the reviewer for raising this important point. Accordingly, we have now removed all claims of neuronal integration from the text, and instead, we state that the newborn neurons receive afferent inputs, which is what the data demonstrates (Figure 4). In addition, to strengthen our claim that the newborn neurons acquire mature electrophysiological traits, we increased the number of neurons recorded from the injured brains. The overall claim still remains the same, but now has additional statistical power (Figure 4). Additionally, to understand what types of receptors are involved in the synaptic transmission to the newborn neurons, we have added data that show elimination of the majority of sPSCs when treated with the AMPA receptor blocker, DNQX (now, Figure 4—figure supplement 2).

Specific comments:

It is interesting to note that the process of regeneration first involves large-scale tissue remodelling akin to 'wound repair' before cell proliferation and differentiation. The authors also show that there is no change in the numbers of cells in the lateral pallium and no cell division either. Does this process then result in re/disorganisation of the lateral pallium? Tracings from the olfactory bulb suggest that this might be the case. Could the authors please comment on this?

This is an interesting point and we have now added new data (Figure 6—figure supplement 1) to answer this question. Immunohistochemistry against NeuN and Tle4, as well as in situ hybridization against *Rorb*, show that the topography of neurons and neuronal populations is properly organized in the injured lateral pallium at 11wpi, compared to controls (uninjured and contralateral). These results suggest that the neurons in the lateral pallium maintain the original organization, but no longer extend their axons to the olfactory bulb (Figure 6). One possible explanation for the lack of OB projections from the lateral pallium of injured (regenerated) brains is that this axonal tract may be severed upon injury to the dorsal pallium. The data thus far suggest that the wound closes first, followed by a process of amplification of cells (likely progenitors, but possibly other cell types) that occurs right at the injury side, before differentiation starts. We have now better discussed this possible model and compared it to the situation in the limb, where the wound epithelium closes and triggers generation of an underlying blastema.

Have the authors at all pursued what the progenitors of the regenerating neurons might be? Are there stem cells that are reactivated upon injury? Or is there a process akin to transdifferentiation that contributes to the regeneration? These aspects could also be discussed in the revised manuscript.

This is a key question, and one that certainly merits investigation in future years. Ependymoglia cells with radial glia-like morphology (and progenitor plasticity) are a likely source for regenerated neurons. However, differentiated endogenous cells of other types may also play a role. Answering this question will require generation of genetically modified axolotl Cre lines that reliably label only selected cell types and thus enable lineage fate mapping/tracing. We have now directly discussed this important point in the revised manuscript, highlighting also the intriguing possibility that reprogramming of endogenous, differentiated cells types (and not only progenitors) may be a possible source, as observed, for example, in the zebrafish retina.